# Multilevel Central Trust Management Approach for Task Scheduling on IoT-Based Mobile Cloud Computing

**DOI:** 10.3390/s22010108

**Published:** 2021-12-24

**Authors:** Abid Ali, Muhammad Munawar Iqbal, Harun Jamil, Habib Akbar, Ammar Muthanna, Meryem Ammi, Maha M. Althobaiti

**Affiliations:** 1Department of Computer Science, The University of Engineering and Technology, Taxila 47080, Pakistan; abidali.hzr@gmail.com (A.A.); munwariq@gmail.com (M.M.I.); habibakbar@uoh.edu.pk (H.A.); 2Department of Computer Science, Govt Akhtar Nawaz Khan (Shaheed) Degree College KTS, Haripur 22620, Pakistan; 3Department of Electronic Engineering, Jeju National University, Jejusi 63243, Jeju-do, Korea; harunjamil@hotmail.com; 4Department of Telecommunication Networks and Data Transmission, The Bonch-Bruevich Saint-Petersburg State University of Telecommunications, 193232 Saint Petersburg, Russia; ammarexpress@gmail.com; 5Department of Computer Science, RUDN University, Peoples’ Friendship University of Russia, 6 Miklukho-Maklaya Str., 117198 Moscow, Russia; 6Department of Computational Sciences, Naif Arab University for Security Sciences, Riyadh 13216, Saudi Arabia; mammi@nauss.edu.sa; 7Department of Computer Science, College of Computing and Information Technology, Taif University, P.O. Box 11099, Taif 21944, Saudi Arabia

**Keywords:** index terms—mobile cloud computing, task scheduling, trust development, energy optimization

## Abstract

With the increasing number of mobile devices and IoT devices across a wide range of real-life applications, our mobile cloud computing devices will not cope with this growing number of audiences soon, which implies and demands the need to shift to fog computing. Task scheduling is one of the most demanding scopes after the trust computation inside the trustable nodes. The mobile devices and IoT devices transfer the resource-intensive tasks towards mobile cloud computing. Some tasks are resource-intensive and not trustable to allocate to the mobile cloud computing resources. This consequently gives rise to trust evaluation and data sync-up of devices joining and leaving the network. The resources are more intensive for cloud computing and mobile cloud computing. Time, energy, and resources are wasted due to the nontrustable nodes. This research article proposes a multilevel trust enhancement approach for efficient task scheduling in mobile cloud environments. We first calculate the trustable tasks needed to offload towards the mobile cloud computing. Then, an efficient and dynamic scheduler is added to enhance the task scheduling after trust computation using social and environmental trust computation techniques. To improve the time and energy efficiency of IoT and mobile devices using the proposed technique, the energy computation and time request computation are compared with the existing methods from literature, which identified improvements in the results. Our proposed approach is centralized to tackle constant SyncUPs of incoming devices’ trust values with mobile cloud computing. With the benefits of mobile cloud computing, the centralized data distribution method is a positive approach.

## 1. Introduction

Internet of Things (IoT) and mobile cloud computing (MCC) are names given to the emerging concept of establishing a meaningful relationship between the actual objects around us to interrelated things that collectively change our traditional lifestyle with progressive and forward-looking ideas [1]. Smart homes, smart cities, and intelligent transportation systems are practical examples of these ideas, where devices and man’s real-time performance integrate to obtain intelligent access to physical changes in our surroundings [2]. It is the biggest revolution of the 21st century and allows every entity of the natural world to be connected, whether it is a group of people, machines, sensors/actuators, or anything else [3]. Therefore, we can say that it is just a name for our ease, through which we can obtain connection to the internet regardless of our location, expensive connectivity devices, and weak Wi-Fi signal issues [4,5]. MCC provides endless connectivity to the IoT and mobile devices to connect and perform their required tasks with less power consumption and less time stamp. Task scheduling for mobile cloud computing enhances sensor processing in IoT devices and mobile device tasks. We investigated and provided significant task scheduling for mobile and IoT-based sensor devices through mobile cloud computing [6].

Mobile cloud is a network of networks where physical objects such as mobile devices and sensor-equipped devices are connected to process their running tasks and update real-time data [7]. The sensors with mobile cloud network (MCN) act as global architecture for advanced level services connectivity, i.e., virtual and physical, information society, and interoperable ICT [8]. IoT allows communication between different heterogeneous devices by the integration of various technologies, i.e., radio frequency identification (RFID) [9], near field communication (NFC) [10], wireless sensor networks (WSN) [11], and mobile cloud computing (MCC). Thus, we can say it is the technology that allows networked devices to interchange information and perform desired activities without manual assistance. Figure 1 depicts the IoT environment with all the devices to communicate [12].

Secure communication among IoT and mobile devices in MCC is not possible without trust evaluation. When two persons come into a relationship, the very first thing developed at that instance is “trust.” Their relationship quality is directly proportional to the degree of trust [14]. The more they trust, the more they stay in contact with one another. Similarly, connectivity is a relationship between mobile networked devices [15]. Figure 2 shows IoT and mobile devices’ trust working to communication after establishing trust values. To make this connectivity robust and reliable, we must introduce trust among mobile devices, sensors, and actuators. Trust motivates the collaboration between two communicating parties. Belief is a single word that defines trust straightforwardly and concisely. Trust can be used in a different context, so everybody defines this term differently to describe the degree of vigorous confidence in someone’s reliability, honesty, and truthfulness [16].

Trust inside the global mobile useability predicts the other mobile device’s behavior. It is a directional relationship between trustor and trustee. The node which desires to communicate with the other party is called the trustor. The other node with which the trustor communicates is called the trustee. Digital trust evaluates past behavior or evidence of the behavior of a device concerning its self-defined level of trustworthiness, which helps perceive its upcoming activities [17]. It is a presupposition to enhance the decision-making for successful cooperation between two agents. Trust between devices occurs at first glance, and it seems very unusual and extraordinary that devices will be expected to show trustworthiness to one another. However, they neither have an intelligence quotient nor an emotional quotient. Figure 3 depicts the relationship between the trustor and trustee devices [18].

Today in IoT and MCC scenarios, many heterogeneous devices communicate with one other. The heterogeneity of devices arises from security, privacy, mobility, power consumption, interoperability, artificial intelligence (AI) adoption, trust, task scheduling, cloud computing, and real-time data processing. Trust between two communicating devices is one of the fundamental issues which must be resolved first. Otherwise, the reason for IoT adoption in MCC will become meaningless. This fact has opened a new research door called “TRUST Enables task Scheduling in MCC” to upload the mobile device’s tasks using MCC. When we talk about social networks, one of the users’ widespread problems is “privacy”. No-one is willing to compromise on security or privacy issues. Most users demand the safe conduct of their confidential data while using online services [19].

Mobile cloud computing (MCC) acts as an alternative to the cloud to compute, store, control, and maintain the network near IoT and mobile devices. It is simply a layer between edge and cloud to process the data before sending it to the actual cloud [20]. MCC makes cloud services more effective as it reduces latency, saves bandwidth and storage, and enhances the quality of service (QoS), while reducing power or energy consumption and CPU time utilization. MCC can connect several mobiles and IoT devices and share services and computational resources among those devices on an on-demand basis. These services are accessible through the third-party platform to connect and effectively collaborate among users. MCC structure is demonstrated in Figure 4. Mobile devices connected from remote locations try to connect with clouds to access the cloud services. The local mobile network provider (MNP) initially accepts all the incoming requests from the mobile devices. The MNP contains a central server and database to provide the services to these mobile users for the mobile network [21]; after obtaining the request, it is prepared for internet service providers to control and coordinate for services. After services are not found on the local server, these services are searched from a remote cloud provider with an effective platform [22,23,24]. In 2012, CISCO suggested MCC to eliminate the shortcomings of cloud computing in IoT and mobile networks.

Users can enjoy three essential MCC services regardless of their physical presence, background, and powerful computing hardware. They can also use the software running on cloud infrastructure (SaaS) [26]. They can even create their application software using different tools, programming languages, libraries, and several other services to access and control only their own deployed applications on the cloud (PaaS) [27]. Moreover, they can use fundamental storage, network, and computing resources in the area where their application is deployed or running (IaaS).

Trust evaluation and modeling are used among mobile devices and mobile cloud servers to estimate the devices’ reliability. The trust improves the cloud computing performance efficiency and enhances secure communication among trustable devices. A model describes whether it computes the trustworthiness of nodes or data. During the modeling phase, it is necessary to decide whether a node’s trustworthiness must be checked or the data’s trust. In [28], the trust modeling among the MCC task schedule was introduced to only offload the tasks which are trustful. These tasks are trustworthy to enhance the trustworthiness of the devices and other related features. Figure 5 shows the trust model in MCC [29]. Trust management is a service mechanism that self-organizes items based on using their trust status to decide. Trust management constructs a framework where mobile devices and MCC draw closer and form a trust-based relationship to exchange sensitive data confidently to process in the MCC virtual machines [30]. This can be done by analyzing and computing the degree of trust in their relationship to make better decisions.

IoT and mobile devices tasks must be scheduled through MCC due to their energy and time constraints. During task scheduling, trustworthiness is one of the important elements because we need to offload only those trustworthiness tasks. This research article focused on this problem faced by the MCC during task offloading. Trust is required to offload the tasks because they execute MCC. The main contributions address time, packet delivery ratio, trustworthiness, and power consumption. The main contributions are the main objectives to adopt in trustable task scheduling in mobile cloud computing through organized algorithms.The proposed technique predicts trustable task scheduling to enhance the efficiency of the proposed system.Task scheduler updates from trustee and trustor to communicate with each other to exchange trust boundaries and then decides through trust computational algorithm for dynamic decision-making.Dynamic trust manager uses trust-based certification to execute and offload only trusted tasks passed from trusted computational models.Trust evaluation and development are handled through Algorithm 1, and correspondence and addition of new mobile node for trust evaluation is checked through Algorithm 2.Trustable task offloading through Algorithms 3 and 4 effectively offloads the task through effective decision-making.We effectively enhance the quality of service (QoS) through a multilevel central trust management approach for task scheduling on IoT-based MCC.Finally, to evaluate the system performance, we analyze the results using mobile offloading through simulation. Our proposed technique indicates that the trust development algorithm and task offloading decision algorithm effectively improves the system decision-making, and less power is consumed through the proposed approach.

Section 2 presents the related work on trust development, task scheduling, and fault tolerance. Then, in Section 3, we present the proposed model for the relevant problem presented in Section 1, using mathematical problem formulation and algorithms supported by methodology and flow diagrams. Section 4 presents a simulation environment using the hybrid approach for task scheduling and problem formulation. Section 5 presents the conclusion supported by future directions for better task scheduling.

## 2. Related Work

According to FCR [12], the cloud is a traditional central server that facilitates almost every customer type by providing ample storage, computations, and network services at very cheap rates. This low cost motivates many users to leverage cloud computing. These are the good aspects of cloud computing. Still, for a moment, focusing on the other side of the picture, it raises some serious problems such as latency, low bandwidth, security/privacy threats, and unnecessary power consumption and time used by the computational resources [31]. To eliminate these shortcomings of cloud computing, we can adopt MCC. MCC acts as an alternate to the cloud to compute, store, control, and maintain the network near mobile and IoT devices. MCC is a nontrivial extension that reduces cloud computing limitations by introducing these features [32,33].

MCC deployment near the mobile devices and IoT layer is responsible for low latency, which is the essential requirement of gaming, video streaming, and augmented reality in mobile and IoT devices. Figure 5 indicates IoT devices with interaction of MCC. A wide geographical distribution of mobile nodes maintains location awareness. This feature reduces mobility issues. The sovereignty of wireless access plays a beneficial role in implementing smart grids and vehicular networks. It introduces real-time interactions as compared to batch processing and the interoperability of the nodes. It gathers the environmental information by negotiating with the sensor-equipped devices (data collectors) and responds to a specific situation using actuators.

Trust is the ability to predict the behavior of another party. The establishment of trust necessitates two or more communicating parties. Trust can be calculated by a multileveled investigation of relationships in different contexts. Otherwise, it may increase additional complexities while increasing the interaction domain [35,36]. Research on multilevel trust, i.e., interorganizational trust, has been minimal compared to individual-level research, and badly affects trust interpretation. After the trust’s calculations, only trustable mobile and IoT tasks are ready to upload on the MCC environment.

The trust computation for IoT and mobile devices are discussed and presented in different research areas. The research conducted effectively defines different proposed approaches to enhance the efficiency of task scheduling in MCC. According to [37], planning, commitment, execution, and integration are the significant steps to be taken before trust at multilevel develops from leaders to administration. Reference [38] discusses the relationship between control and confidence and concludes its dependency on institution and situation. The role of collective trust is later examined, and individuals’ reactions to changes is investigated. The theoretical and illustrative effect of individuals in shaping their organization has been investigated. In [39], the authors suggested the need for differentiation between trust and distrust. An overall consideration of temporal dynamics is vital as exchange relationships can be changed and affect trust. Lastly, fluidity between people and the environment has been called upon to be explored; refs. [40,41] came up with the idea that the relationship between people and place is elastic and can be worked upon after understanding multilevel trusts. The trust computation is directly linked with MCC. After trust, the next work is to upload the mobile cloud for processing on virtual machines (VM).

After trust computation, the literature is moving towards multilevel trust management frameworks for service-oriented environments. According to [42], IC3 reported a 22.3 percent increase in online fraud to provide trustable MCC services, which is a big reason to distrust online services. Running online businesses demands numerous vendors’ and consumers’ requirements according to their role to control and provide MCC services to the end-users. Trust is the basic need of every business, whether online or traditionally [43]. Trust in any situation is directly related to the certainty of risk. If the risk is higher than the trust, it is nonsatisfactory [44]. In an open environment of e-commerce, feedback from participating parties plays an important role. So, detection of falsified feedback is necessary. Falsified feedback can make honest participants incredulous, and dishonesty can be considered dubious. The collected feedback can be investigated to accurately picture predicted risks and make consumers feel more confident in online services [45]. This paper proposes a multilevel trust management framework for task scheduling in the MCC environment to enhance the task offloading efficiency of mobile and IoT devices under the domain of security.

Gupta et al. [46] provide knowledge about on-demand internet access using cloud computing models and ubiquitous computing resources. This model gained popularity to support such models. Moreover, the NIST cloud explains MCC’s multiple distinguishing features: rapid elasticity, measured self-services, on-demand services, broader network access, and rapid elasticity. In MCC, the resources are distributed over the network, and distribution provides heterogeneity among resource-sucking devices. Trustable resources are not effectively utilized for efficient workflow and effective features distributions. Fault tolerance is significantly overcome due to trustable communication in MCC. Liu et al. [47] checked the resource distributed with fault tolerance and millions of mobile devices disconnected from any service disasters.

The recent technique for task scheduling in MCC is presented in Table 1. We compare the recent literature for task scheduling in MCC for fault rate, energy optimization, heterogeneity, storage, time, task offloading, control message, and percentage of tasks to be offloaded. The parameters are discussed for IoT and mobile devices to be offloaded to the mobile cloud or executed on the mobile device [48,49]. Table 1 effectively compares the related results obtained through these proposed techniques. Table 1 shows that fault rate, energy optimization, time constraints, and offloading are either not evaluated collectively or are evaluated with lower results than the proposed technique.

## 3. Methodology

### 3.1. Model Structure

Today, almost everybody is becoming part of social networks. Still, no one is willing to compromise on privacy, one of the most common problems with IoT deployment. The lack of trust plays a vital role in hindering using online services to deal with this problem. We propose developing an environment where users can feel confident to upload the mobile- and IoT-based devices to MCC. This approach works in two ways. First, it computes the trustworthiness of every device and task. The trustful functions should be distributed to the cloud layer and provide the significance of controlling all the automation techniques. Figure 6 depicts the working of task schedular to support for experience based trust computation and trust evaluation. 

The model introduces a way to accomplish a trustworthy interaction between two communicating nodes, i.e., mobile/IoT devices and MCC. The other nodes that allow them to communicate with it are termed the trustor, while the other communicating participant is the trustee. Whenever a new trustee joins a network, its behavior is unpredictable before its positive or negative performance. After starting a communication, the MCC recommends the trustee participate in the network for some specific period under controlled access. After completing the first communication session, the trustee’s performance explains why the trustor can evaluate the trustee’s trustworthiness through its experience. The mobile cloud layer controls and provides full access and performs all the tasks’ computations. Reliability can be formulated by considering trust properties such as honesty, latency, reluctance, and competence. All properties are evaluated individually to give readings in numeric values. Figure 7 depicts the computation of the trust of the devices and the task scheduler schedule for the MCC tasks.

All values collectively give wholesome weight by performing some statistical operations, and that value is the trustee’s trust value. After this calculation, the trustor sends the weight toward the task scheduler to update the trust value based on the threshold. If it has any previous value, the trust values will be updated by simply taking previous and current values. The resultant values are sent to the MCC server. These values are broadcast to all the nodes in the network. This strategy may also contribute to resolving mobility issues and the quality of services. The only trusted tasks from the devices towards the cloud are shifted and provided through trust propagation. All values collectively give wholesome weight by performing some statistical operations, and that value is the trustee’s trust value. This strategy may also contribute to the resolution of mobility issues and the quality of services. The working of the task scheduler is enhanced with time as the system becomes mature. The proposed approach builds on the top of the trust levels and provides the significance of these values. The task scheduler schedules only those tasks that passed the trustor and trustee values’ IC and CT criteria. When a trustee is an old participant in the network with a trust level history, it has those trust values labeled. Those labels serve as a pass for a trustee to communicate on the web. These values range between 0 to 100, which serves as a scale of trust value to schedule MCC for task processing. Figure 6 elaborates the trust computation with the task scheduler, using experience-based trust evaluation.

If the label on any trustee is less than 50, it can only process on the local machines, i.e., mobile devices and IoT devices.

If the trust value exceeds 50, it schedules through the scheduler towards the MCC.

Trustees with a trust value of more than 90 can be selected as direct service providers to the MCC through direct task scheduling.

In the situations in which a trustee is to be taken as a service provider by any node, it is required to be a trustful entity. Its label is matched with the centrally propagated trust value to avoid any misleading circumstances. If its value fulfills the threshold requirement, the communication starts; otherwise, the access is denied.

After completing the first communication session, the trustor computes the trust values based on its experiences with the trustee. Two different aspects of the trustee are considered to evaluate its trustworthiness in two different levels. In level 1—social trust, Liu’s technique [16] is used to determine whether the device is honest or not. If the device comes out, to be frank, in level 1, it goes to level 2 (QoS trust) for further evaluation. At this level, the assessment considers availability and reliability as two standard trust properties. After passing through these trust levels, the task is ready to schedule for MCC’s uploaded server. MCC is a directory connected to cloud computing to gather some of the services that are required to process. This technique is novel to provide efficient trust management entities and significantly enhance the trustworthiness of these entities. Figure 8 depicts the central model diagram from the trust development and task scheduling for mobile cloud computing. Algorithm 1 discusses and computes QoS trust evaluation and development. Trust development enhances the selection of trustable tasks from mobile and IoT-based devices. The output of Algorithm 1 is the trust computation of the new functions from mobile and IoT devices.
*Trust_evaluation*(*i*←*j*)(1)

Equation (1) computes the trust evaluation for the trustor *J* and trustee *I*. After the trust evaluation, the trust identification is performed and provides the significance to control the trust management.
*Check*: *I*←*j* (*ID* + *L_i_*)(2)

Equation (2) defines and check the trust from trustee to trustor. The *ID* and *Li* are the required parameters to compute the trust values. Values from trustor *I* and trustee *J* are based on the new trust computation (Algorithm 2) or directly added to the social trust adoption technique (Algorithm 3).
*T_avi_*: *I* ← *j* (*avi*_*i*←*j*_,*rel*_*i*←*j*_)(3)

Equation (3) defines the trustable nodes to be selected for the final computation. The trustee devices enhance the probability of the nodes and significantly reduce the task offloading phenomenon. In Algorithm 3, we adopt the social trust adaptation technique to compute the reliable social level of trust among all the different devices’ tasks. The adaptation technique is taken from research to calculate the vehicle’s total confidence to control the system’s operations with an efficient control view and control mechanism. Social trust is adopted for every node, and every node is responsible for adapting these trust values for efficient resources. If the adaptation technique is not followed, the task is refused access to upload to the cloud server. The jumping from one algorithm towards another makes task scheduling efficient and trustable. Algorithm 4 is used for the trust adaptation and trust computation technique to schedule the MCC tasks. All the information from job nodes and IoT devices is fetched to control the job descriptions. After they finalize the nodes and selection procedure, the nodes are sent back towards the scheduling. After the trust computation, these nodes were completed, and fulfilled the criteria to control these specifications. The main scenario to schedule the task is based on specific parameters to be fulfilled at this task scheduling stage.
(4)$$F\;\leftarrow \;\frac{∆T_{m}}{∆T_{exc}}$$

In Equation (5), the processing time is computed, and this time rivals the total execution time of the trustable task from the mobile or IoT devices. The job threshold size, which is equal to the job execution time and execution adaptation technique, is similar to the job description time and job running time.
*Job_exe**_M_*()(5)
*activeCloud*_(*VM*)_(6)
*submit*_*C*(*J*)_()(7)
*exe_job_*()(8)

**Algorithm 1.** QoS Trust Evaluation and Development.**Input:** Mobile Nodes, Sensors, and IoT Devices**Output:** Trust Validate1:     *trust_evolution*(*I* ← *j*)2:     *j^ID^*            //            Trust Identification3:     *J*_*req*←*I*_       // Send Request to Trustee *i*4:     check: *I* ← *j* (*ID* +*L_i_*)5:     if (*j*! = *L_i_*)           Go to Algorithm 2    else           go to Algorithm 3    end if6: *T_avi_*: *i*←*j*(*avi* _*I* ← *j*_, *rel* _*I* ← *j*_)                      // Availability and Reliability7: *T_f_*: *T_avi*: *i*←*j*(*avi*_*i* ← *j*_, *rel*_*i* ← *j*_) 8: if (*T_f_* > 90%)            service_provider (*T*_*i* ← *j*_)        else if (*T_f_* > 50% && *T_f_* < 90%)                   network_comm (*T*_*i* ← *j*_)              else                   dumble_terminal (*T*_*i* ← *j*_)      end if  9: trust _*I* ← *j*_ ( ) ← published ()

**Algorithm 2.** New Trustee.**Input:** New Node(Mobile Device, Sensor, IoT De            vice)**Output:** Trustable new Entered Node2: **Start**1: *j*_(*i*)_2: **if**( *j*! = *L_i_*)         permission_grant(*F_n_*)         go to Algorithm 1 step 8    **else**3: Go to Algorithm 34: **End**

**Algorithm 3.** Social Trust Adaptation Technique.**Input:** Nodes (trustor, Trustee, Adaptation)**Output:** Calculated Social Trust1: **Start**2: social_trust(*I* ← *j*)3: *j*_(*i*)_4: **if**(*j*! = *h_r_*)                // Checked through adaptation technique [X. Liu, et al. [16]]           *request_refuse*()5: **else**            Go to Algorithm 1 step 86: *social_trust*_*calculate*( )_7: End

**Algorithm 4.** Task Scheduling Decision.**Input:** Input from Table 1 (LEGENDS Table)**Output:** Job Scheduling 1: MobInfo(*B*, *T*, *L*, *App*, *S*)2: JobNum(*m*)3: NodeNum(*n*)4: FetchInfo(*T*)5: CreateNodeNew(*VM*, *N*, *Schedular*)6: Execution of Algorithms 1–3.6: Send_(*T*, *D*, *C*)_ ←
*Schedular*( )7: for (*T* ≥ 0) do            *calculate*_*exe_time*( )_                 *F* ← Δ*T_m_*/Δ*T_exc_*    end for8: while (job_size ≤ threshold) do             *C*(*B*, *F*, *M_b_*, *M_loc_*, *M_storage_*)            if (*C* ≤ *M*) then                   *job_exe*_*M*_( )            else                 *activeCloud*(*VM*)                  *submit*_*C*(*J*)_( )                  *exe*_*job*_( )          end if     end while9: *Job_state*_*Store*_( )

### 3.2. Trust Factors

Trust factors are checked thoroughly at this level. These two are the critical factors used in avionics to estimate the risk of failure and decide on it. The selection of these factors shows their importance, where any wrong choice has significant, real-life consequences in people’s lives. We opted for these features as standards for calculating trust value to cope with vulnerabilities that can cause network failure. Our proposed strategy can develop the fear in users of being penalized in real time by technically controlling their network access based on their behavior. It compels them to behave positively later, if not the first time. Figure 8 shows the complete flow model for trust computation and task scheduling in the proposed technique.

### 3.3. Reliability

Reliability can show satisfactory performance or the possibility of failure of a system. It can be measured through factors that reinforce the validity of a system. We considered the energy-consumed rate, time taken to respond to a request, and packets delivery ratio as evaluation matrix.

### 3.4. Availability

Availability is the measure of unpreparedness of trustees during network communication. It can be measured by estimating the possibility of downtimes in the lifecycle.
(9)$$Trustee_{availability}\;\leftarrow \;\frac{request\_compt\_time()}{depay\left(\right)+requestcompttime()}$$

Equation (9) perform trust availability for new tasks. Both properties are considered individually to give numeric value readings used to produce a wholesome value by performing statistical operations. The resultant value is regarded as the trustee’s current trust value. After this calculation, the trustor sends these values to the fog node to update current trust values. The fog node checks whether it has any previous trust value, then the trust is calculated by simply taking both current and previous trust values. Otherwise, the trustor’s direct observation is considered trust value (this only happens when a new node joins a network). This trust value is then sent to the cloud server to broadcast it for all fog nodes and label it to the device. This label is added to profile information of the device to be available for the time of its next confrontation with other network devices. This way, the impact of the trustee’s previous behavior in the form of trust value is publicly visible, and its reputation proceeds it. This approach can also be used to eliminate mobility issues in IoT devices.

## 4. Results and Discussion

We implemented the algorithm in MATLAB to evaluate our proposed model. MATLAB is one of the best simulators for MCC simulation, specifically for trust computations and energy requirements. The scheduling experiments were conducted on Core i5-CPU/3.0 GHz/8 GB RAM-running PCs and MATLAB R2018b. We design a user-friendly and creative environment using MATLAB to build our experimental model [1]. The experimental model is designed to achieve the reliable and efficient behavior of IoT devices. Two trust parameters are selected; the first is the “availability”, and the second is “reliability”, which is already discussed in the previous section of the proposed work. We evaluated our work by comparing it with Greedy Perimeter Stateless Routing (GPSR) [4]. We established an IoT environment with 100 edge nodes (*T_e_* and *T_r_*), 10 MCC NodesFn, and a mobile cloud server. The evaluation matrices used to measure the trustworthiness of a device are as follows.

### 4.1. Time

Proper time management helps us accomplish the top job in the minimum time. Our model performs multiple communicational and computational tasks simultaneously in less time than GPRS. Equation (10) computes total time from task submission to completion. Figure 9 shows the time to manage Trust Request, Trust Development, Trust Upgradation, and Propagation by comparing methodologies with proposed technique.
(10)$$T_{total}=\sum_{req}^{comp}T\left(T_{e}+T_{r}+F_{N}+C_{s}\right)$$

Total is the total time consumed by our model. T is the time taken by the trustee (*T_e_*), the trustor (*T_r_*), mobile cloud node (*F_N_*), and the cloud server. Figure 10 shows the number of tasks increased to 20,000 to check the validity of the proposed system—the proposed system effectively collects makespan time for effective monitoring and validation of the proposed model. The results also elaborate that our system performs effectively well under such conditions as time passes. In Figure 9, the request elaborates on the request received, and the time it is measured against shows the actual propagated time for request experimentation. T-Development is trust development time. The T-Development time shows the performance of the proposed system, which is better than RGP and MLT techniques. T-Upgradation is trust upgradation. Therefore, trust development time for upgradation of trust enhancement is effective and provides better results than other approaches. Propagation refers to the trust development with upgraded values to show the trust enhancement values for effective analysis and effectiveness. The proposed system shows effective results—overall, 21% better results for GPGR than RGP and MLT approaches.

### 4.2. Packet Delivery Ratio

Measurement of data concealed or dropped by the trustee can predict the trustee’s intentions. When a node shows harmful intentions, it can never be considered reliable. Previous works measured packet delivery in a specific period, but we measured it differently.
(11)$$PDR=\frac{DR}{DR+DL}$$
where *PDR* is packet delivery ratio, *DR* is data received, and *DL* is data lost during scheduling in the proposed technique.

We simulated our proposed technique by deploying 100 mobile or IoT nodes and a central server. Figure 11 shows that our proposed technique achieves high accuracy of value. According to the estimated result ratio, the proposed technique achieves 97% of the packet’s delivery r. Overall packet delivery ratio value is 97.865%, which is more powerful than other techniques.

### 4.3. Energy Consumption

Energy consumption plays an essential role in the success or failure of a model, so it should be measured carefully. Econ is the total energy consumed by a model. E is the energy consumed by request submission (RQ), trust development (TD), trust update (tup), and trust propagation. Again, we are comparing our results with MLT and RGP. Figure 12 shows the MLT and RGP comparison with proposed technique. 

The results are compared with two approaches, i.e., GFMS and SMTE. Both techniques are designed and developed through proposed task scheduling, and an enhanced version is required to provide effective task processing. Initially, the energy level starts with a high peak time, but the energy consumption remains low with time. The proposed technique shows low energy consumption and enhances energy consumption by providing effective and efficient optimization. The proposed technique shows better results than GFMS and SMTE approaches from the literature. Figure 13 shows these results. 

In addition to the time and power consumption, the VM’s availability and migration during task scheduling are the other main contributions. Figure 13 shows the comparison of VMs migration during the task scheduling in the proposed technique compared to DRA and MARKOV analysis. We compared the results on multiple VMs, which shows that during the migration, our proposed technique works better to adjust the VM migration from one platform towards another platform for effective and efficient task scheduling. Time and powers are less consumed in the proposed model whenever scheduling a task needed to migrate from one VM to another. This happens when one VM falls, or any other reason for processing and scheduling tasks.

Experiential-based tasks are amplified in consignment (number of functions become batch) and the calculations are excluded, so the tasks’ power is smaller than other methods. The chance is calculated throughout Equations (1)–(12). Moreover, the computational possibility calculation, shown in Figure 14, demonstrates the most excellent and minuscule offloading likelihood.

Figure 15 shows the trust comparison among both techniques. With time, the expected trust from the trust comparison and real trust values shows the enhanced trust obtained from the proposed technique. These trust values are evaluated from Equations (1)–(8) and Algorithm 3. The trust is computed, and results show effective trust computation with efficient and effective trust enhancements.

They are grounded on the findings of the task’s possibility and statistics of mobile device structures shown all through Figure 15. Indicators such as battery information, storage, offloading time, bandwidth, and job completion rate are powerful task offloading frameworks. The time and power consumption are shown in Figure 16. The figure enhances both parameters with effective time and power management. On the other hand, Figure 17 depicts the results of tasks submitted towards the cloud after the final decision. The decision is purely made based on probability and results obtained after Algorithm 4. The results show that tasks requiring more power, time, and cost must offload towards MCC VM after the computation of trust values computed through Algorithms 2 and 3.

## 5. Conclusions and Future Work

We proposed a centralized multilayered trust management model for task scheduling in MCC to prove our two-layer trust evaluation model with simulations. The results proved to be better than MOGA [10], EETS [11], MECCO [12], GPSR [4], DRA [13], RGR, and MARKOV [15] (previously accepted techniques), demonstrating our selection of availability and reliability as the trustee’s trust evaluation’s exemplary standards. Our selected parameters and our model of centrally synchronized MCC nodes such as mobile devices and IoT devices appear to be the best option for trustable task scheduling. Nontrusted tasks from mobile devices and IoT nodes cannot schedule through the cloud when it can achieve low latency from task submission, centrally synchronizing all fog nodes with trustee trust values, triggered at the interaction of a trustee in a network. The proposed model effectively enhances the results by 20% less than previous techniques from literature, makespan time by 26%, packets delivery and trust computation by 21%, trust values by 17%, and our devices consume 23% less power than the proposed technique.

In the future, the work can consider more security through modern security parameters and provide the significance to handle the resources provided. Additionally, we plan to compute the trust through deep learning and task scheduling through AI constraints.

## Figures and Tables

**Figure 1 sensors-22-00108-f001:**
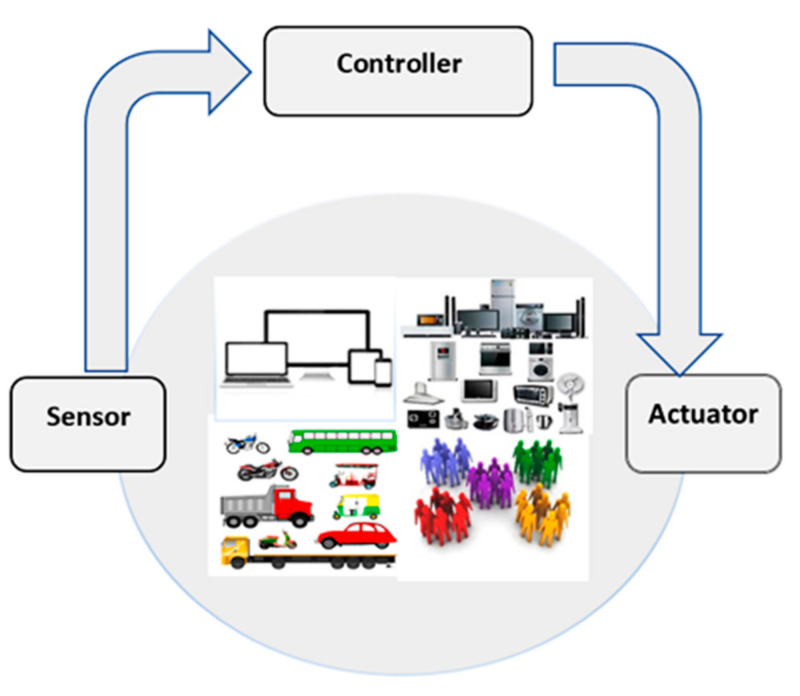
Use of controller, sensors, and actuators in IoT environment [13].

**Figure 2 sensors-22-00108-f002:**
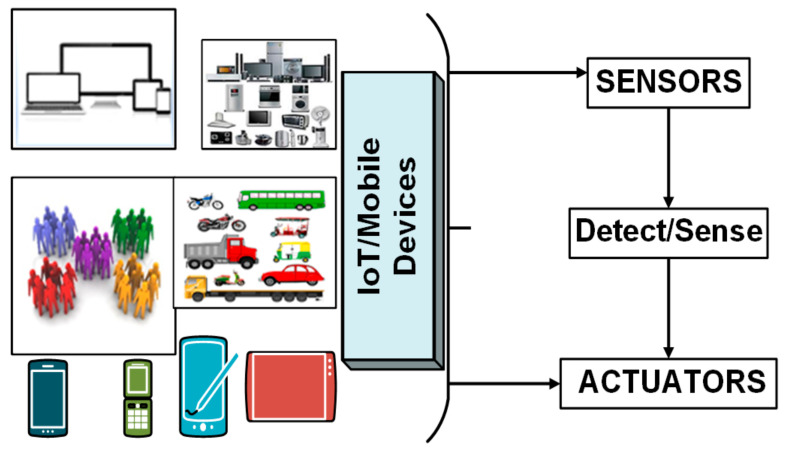
Mobile cloud with IoT and mobile devices for data sensing.

**Figure 3 sensors-22-00108-f003:**
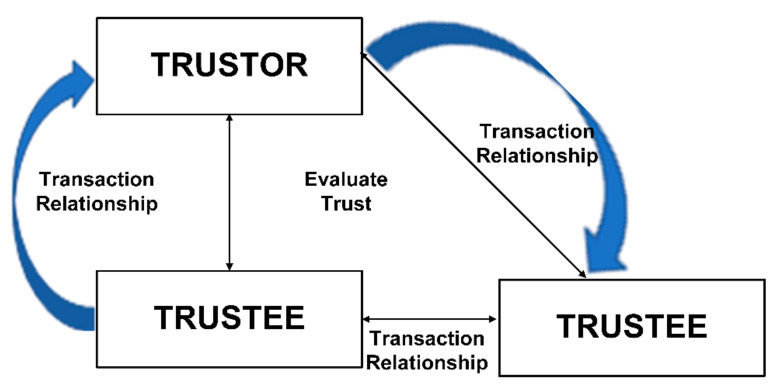
Relationship between trustor and trustee [18].

**Figure 4 sensors-22-00108-f004:**
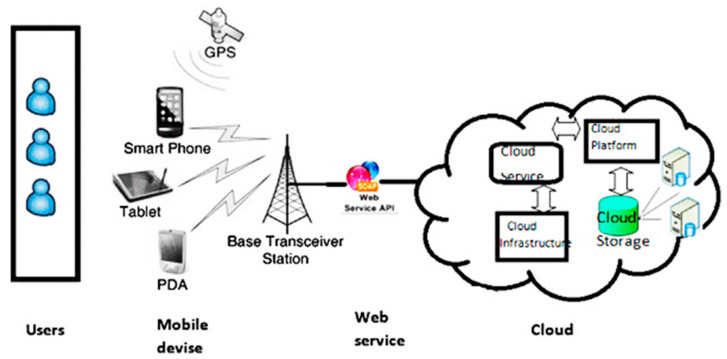
Mobile cloud computing for users, mobile devices, web services, and cloud [25].

**Figure 5 sensors-22-00108-f005:**
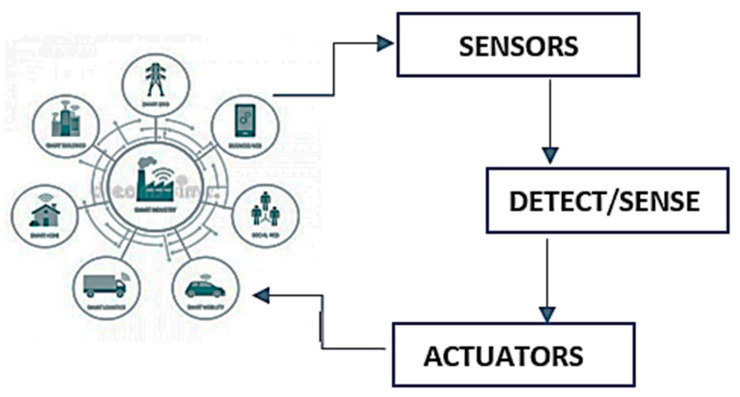
The role of MCC computing in mobile and IoT [34].

**Figure 6 sensors-22-00108-f006:**
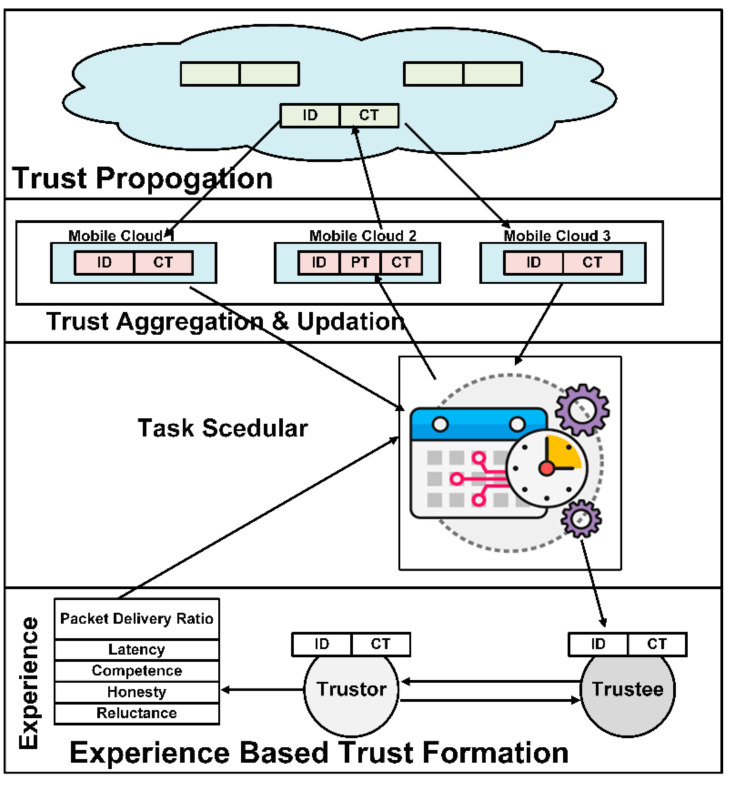
Mobile task scheduling and trust computation.

**Figure 7 sensors-22-00108-f007:**
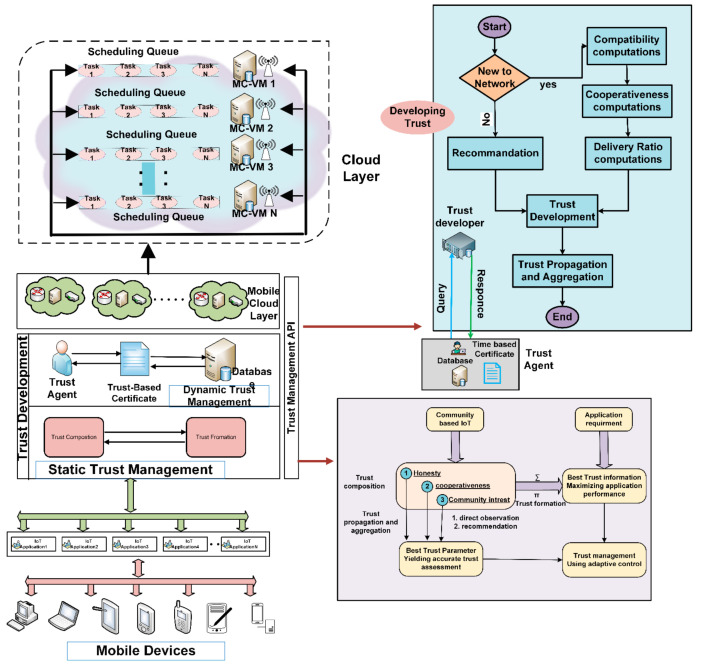
Trust development and task scheduling for mobile cloud computing.

**Figure 8 sensors-22-00108-f008:**
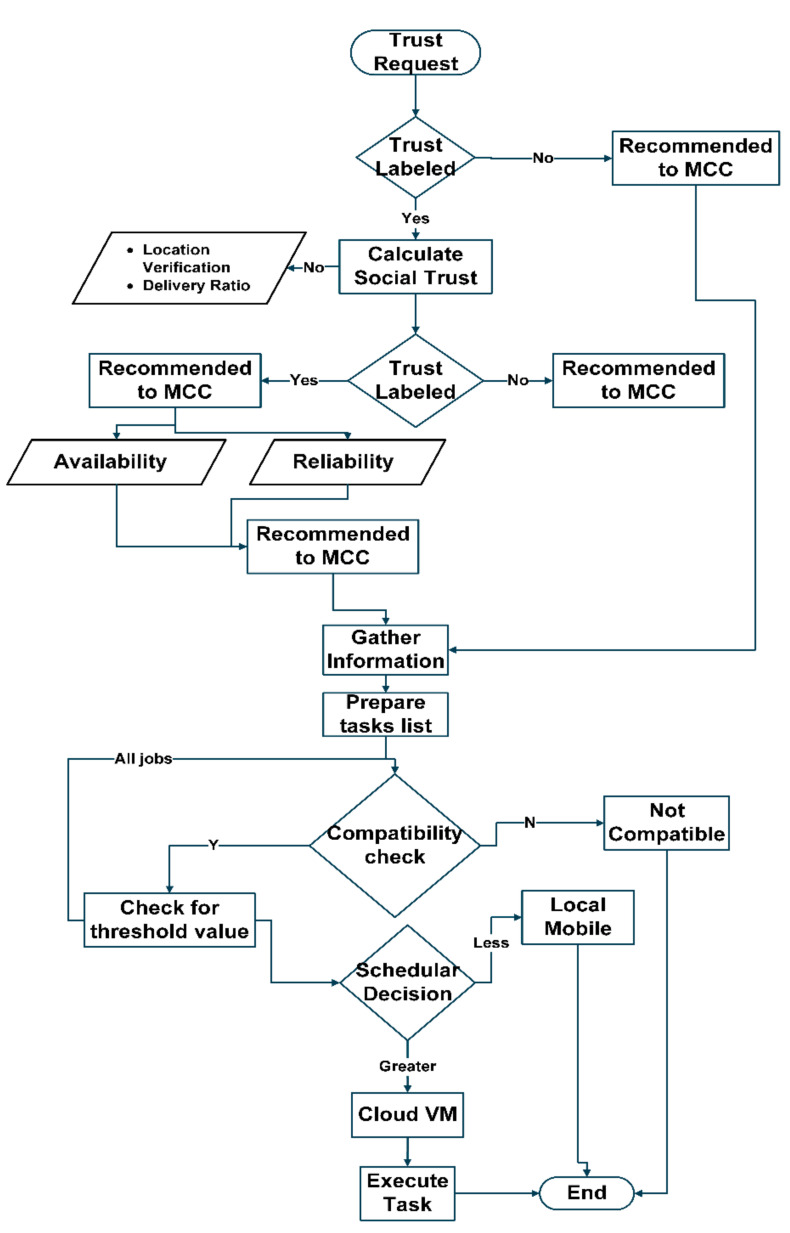
IoT and mobile cloud trust development flow model.

**Figure 9 sensors-22-00108-f009:**
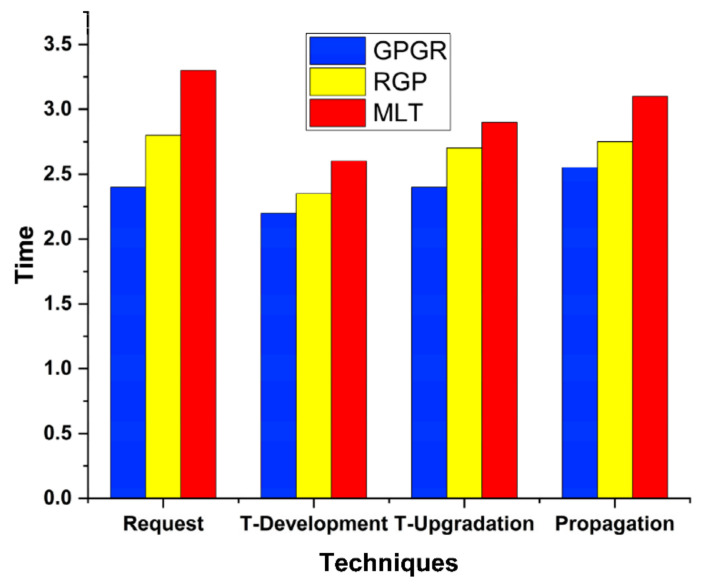
Time for request completion.

**Figure 10 sensors-22-00108-f010:**
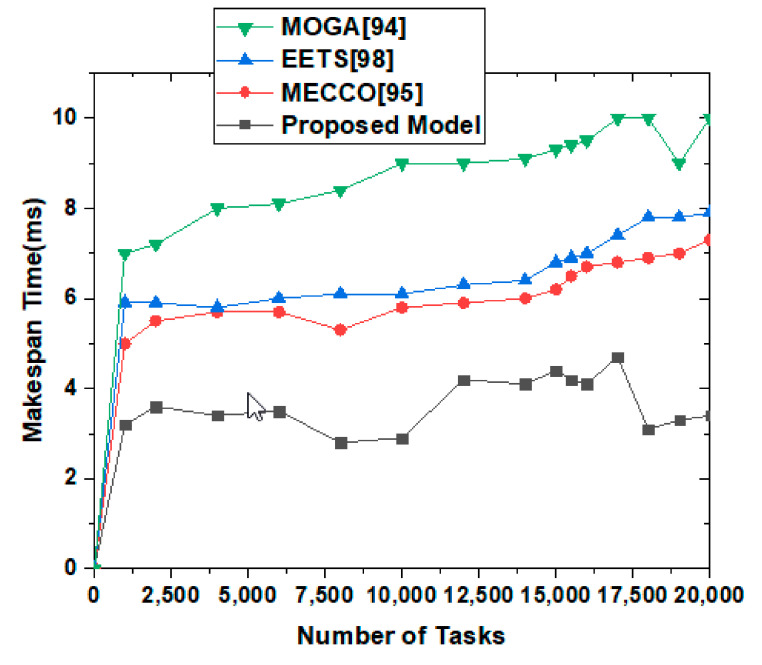
Makespan time consumption after increasing the number of tasks to 20,000.

**Figure 11 sensors-22-00108-f011:**
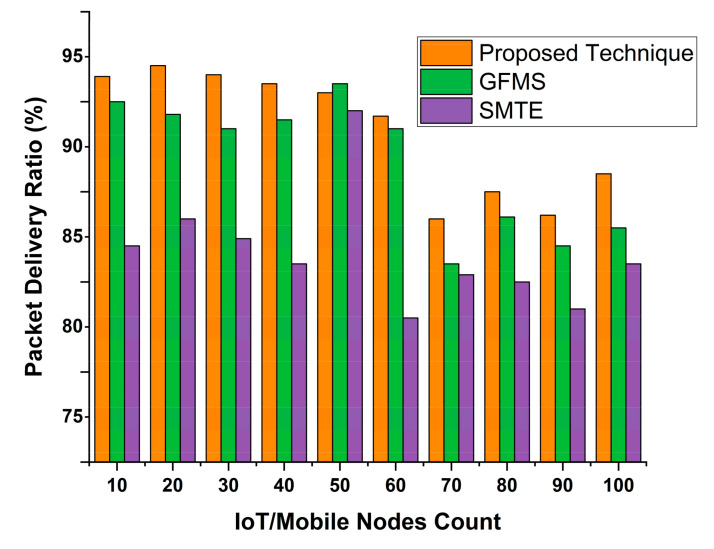
Packet delivery ratio.

**Figure 12 sensors-22-00108-f012:**
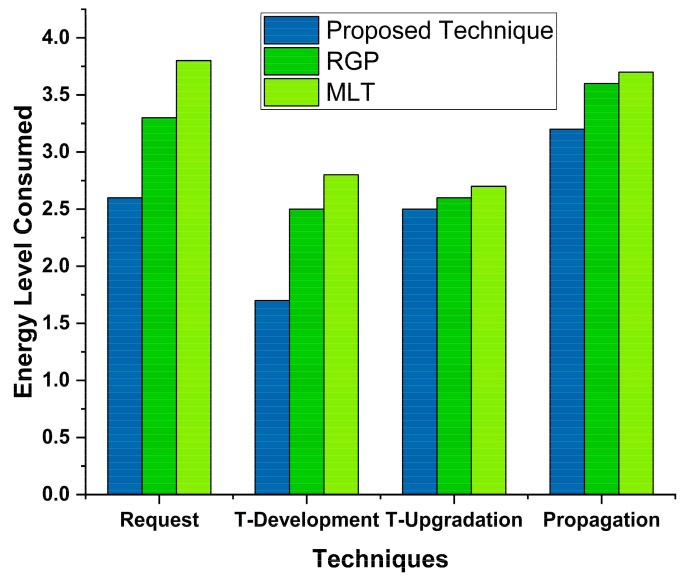
Power consumption while task offloading.

**Figure 13 sensors-22-00108-f013:**
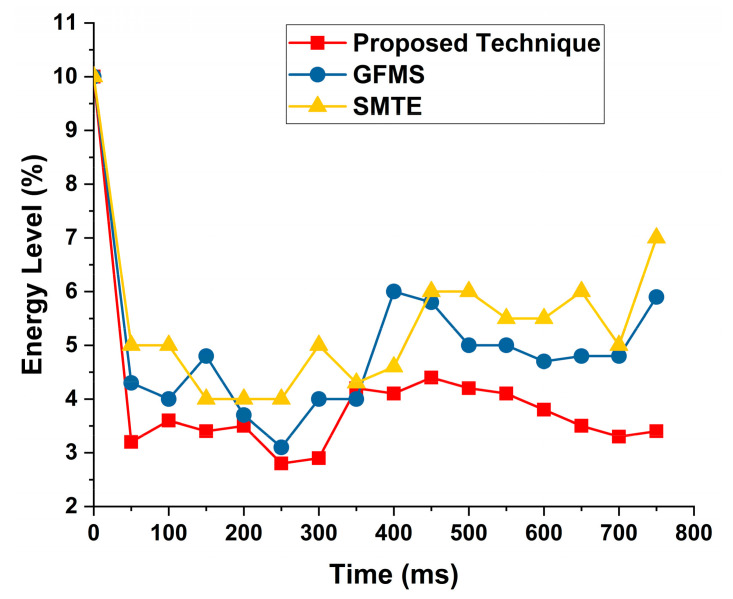
Task energy level consumption while task offloading to MCC VMs.

**Figure 14 sensors-22-00108-f014:**
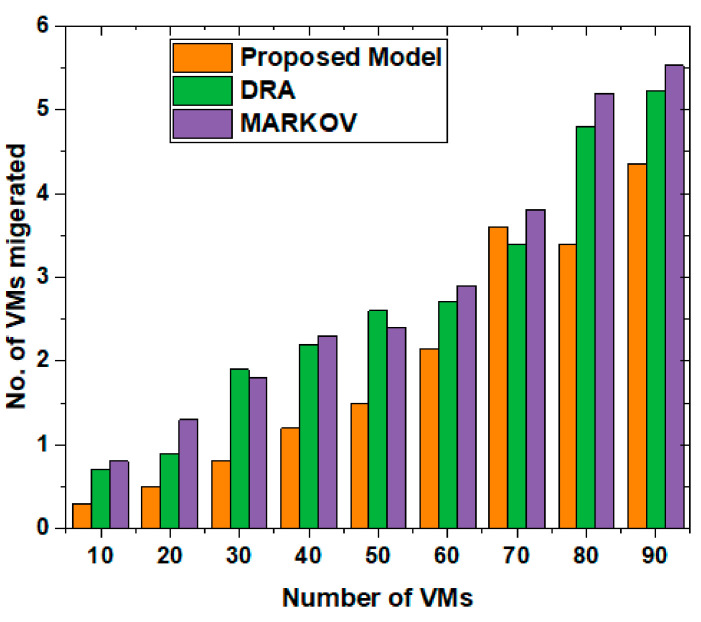
Comparison of VMs for tasks migration.

**Figure 15 sensors-22-00108-f015:**
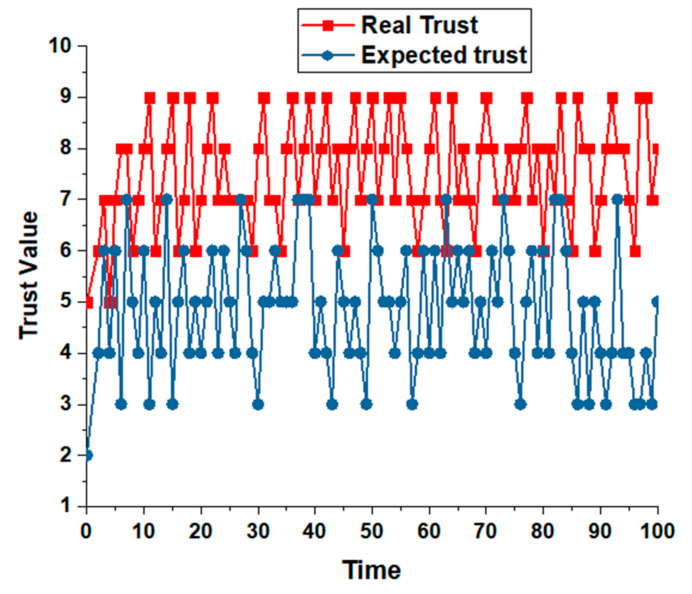
Trust value and trust computations based on time.

**Figure 16 sensors-22-00108-f016:**
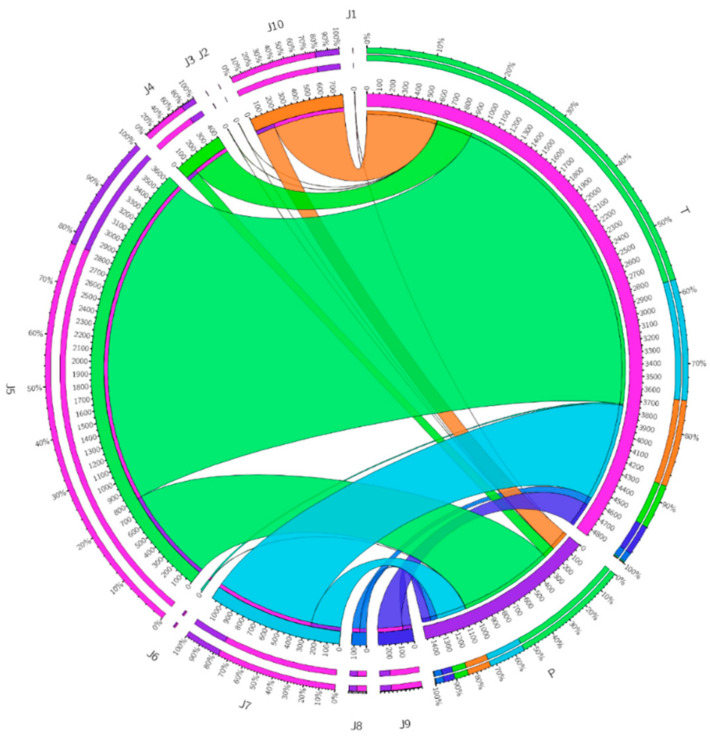
Time and power consumption in the proposed approach.

**Figure 17 sensors-22-00108-f017:**
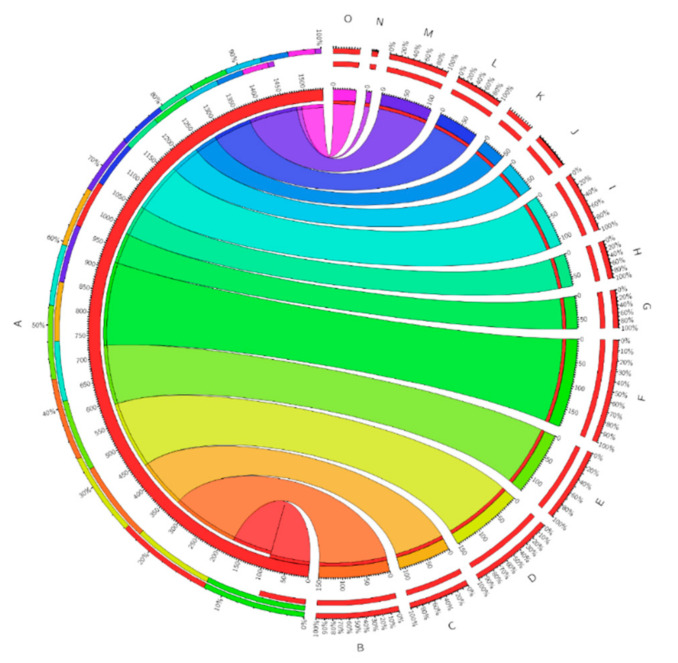
Request submitted to the cloud of the proposed system.

**Table 1 sensors-22-00108-t001:** Comparison of IOT-based task scheduling.

Proposed Papers	Fault Rate	Time	Energy Optimization	Offload	Heterogeneity	Control Messages	Storage	% of Task Executed
Lee et al. [50]	✓	✓	-	✓	✓	✓	-	-
Raju et al. [51]	✓	✓	-	-	✓	-	-	✓
Abd et al. [52]	✓	✓	✓	-	✓	-	-	✓
Park et al. [53]	✓	✓	✓	✓	-	✓	✓	✓
Al-Sayed et al. [54]	✓	✓	-	-	-	-	-	-
Kashanchi et al. [55]	✓	-	-	✓	-	-	✓	-
Peng et al. [56]	✓	-	✓	-	✓	-	✓	-
Tang et al. [57]	✓	-	✓	-	-	-	-	✓
Lin, Xue, et al. [58]	-	-	✓	✓	-	-	-	✓
Guo et al. [59]	-	-	✓	✓	-	✓	-	-
Wei et al. [60]	-	✓	-	-	✓	-	-	-

## Data Availability

Not Applicable.

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
