# Peer review of "Multilevel Central Trust Management Approach for Task Scheduling on IoT-Based Mobile Cloud Computing"

_sensors, 2021, doi:10.3390/s22010108_

Round 1

Reviewer 1 Report

The authors need to take in consideration the following suggestions:

  • The introduction needs to be improved by discussing current articles because only a few new articles have been included in it.
  • The quality of all figures needs to be increased, also the size of labels needs to be increased.
  • A new section named “results and discussion” is necessary in order to condense all the obtained results in this work.
  • The authors need to explain better the experimental setup.
  • The authors do not compare their results with another proposal in the literature recently in order to validate or show that their contribution is important in the subject.
  • The conclusion needs to be improved, adding quantitative results not only qualitative results. In addition, it is important to mention, what is the next with the investigation?

Author Response

Dear Reviewer,

Kind Regards

Reviewer 2 Report

In this paper, the authors propose a multilevel trust enhancement approach for efficient Task Scheduling in Mobile Cloud Environment. Then test the algorithm and get a good results. I agree that the author has made a considerable contribution, the approach also has a very large prospect in the field of Mobile Cloud Computing. But I think there are still some problems that need to be revised to improve:

  1. This paper is not well structured: The main work of this paper should be the proposed credible task scheduling model and algorithm in MCC. However, in the introduction and related work, the authors introduce a lot of irrelevant information and too many of redundant work about the Internet of Things and fog computing, which may be allowed in a review paper.
  2. The contribution of the article is insufficient or unclear: same as 1, the article introduces too few innovations in trust and task scheduling or is not concise enough. Experimental comparative protocols are too old, such as GPSR, which was proposed in 2000. Authors should compare models mentioned in related work, or recently proposed protocols such as EBGR, QNGPSR, or ARdeep to increase the persuasiveness.
  3. There areproblems with the grammar of this article, such as: “cope with this growing number of 19 audiences soon”, “calculated nodes Time”, “GPRS” or “GPSR” or “GPGR”? “RGP” or “RGR”? and so on.
  4. Many pictures, formulas and concepts are not introduced or are wrong: Fig13/14 are not clear, what is “RGP/RGR”, what is “ and ”, the x-axis of Fig10 is wrong.
  5. In section 4 (Experiments and Discussion), Lack of necessary discussion on experimental results.

Author Response

Dear Reviewer, 

Kind Regards.

Reviewer 3 Report

The paper proposes a centralized multi-layered trust management model for Task Scheduling in Mobile Cloud Computing (MCC). Then the proposal is evaluated against two existing techniques.

After having reading the paper a couple of times it is still not entirely clear to me whether the goal is to assign more tasks to the nodes in the MCC that are more efficient and therefore get a higher trust value, or to make a decision about which tasks need to be delegated to the cloud. Besides, I am not entirely convinced of how the presented approach takes into consideration the processing capabilities of the infrastructure. For example, even if all the devices trust a node, the node might get blocked or overloaded at some point if its current load is not considered for the trust computation.

At some point, the paper states “This paper proposes a multilevel trust management framework for task scheduling in the MCC environment on service-oriented marketplaces.” This is quite confusing, and I fail to see the relationship between these marketplaces and task scheduling.

In the related work section, it is sometimes unclear for me to whom the word “researchers” refer to. Besides, in my opinion, some explanation about Table 1 is missing.

In page 8, it is mentioned that “All values collectively give wholesome weight by performing some statistical operations, and that value is the trustee's Trust value.” Which are these statistical operations.

Liu's technique (section 3 page 9) should be explained an cited properly to facilitate the understanding of the algorithms.

In Figure 8, which represents the Flow Model, there are 3 consecutive “Recommended to MCC” stages, is there any reason for that?

It is not clear what represents figure 9, moreover it does not have any units for the time and RGR is not defined. Since the proposal is being evaluated against both RGR and GPGR, I believe that it should be described how these techniques work. Note that according to the figures the proposed technique outperforms them, but the paper does not provide an analysis nor a comment about these results. Additionally, how the propagation time is computed? The paper does not mention how the computed trust values are propagated through the cloud servers and the experiments only seem to consider 1 single server.

In section 4.2 trust is proportional to 1/datahiding. I do not get the meaning of this datahiding value, neither do I get where it comes from or how it is obtained. Further, what is the advantage of using it instead of using packet delivery measurements as previous works?

Maybe other readers are more familiar than me with the diagrams presented in figures 13 and 14, but I would appreciate if a brief description about how to interpret them is provided.

Finally, just a minor comment since I am not used to the concept of trustable tasks, are they the same as trusted tasks? The same goes for schedular, is it scheduler?

Author Response

Dear Reviewer,

Kind Regards

Round 2

Reviewer 1 Report

the authors have correctly addressed the reviewer's concerns. Hence, I can recommend accepting the article.